# Controls of thermal response of temperate lakes to atmospheric warming

Jian Zhou [1,2], Peter R. Leavitt [3,4] ✉, Kevin C. Rose [5], Xiwen Wang[1], Yibo Zhang[1], Kun Shi [1] ✉ & Boqiang Qin [1] ✉

Atmospheric warming heats lakes, but the causes of variation among basins are poorly understood. Here, multi-decadal profiles of water temperatures, trophic state, and local climate from 345 temperate lakes are combined with data on lake geomorphology and watershed characteristics to identify controls of the relative rates of temperature change in water (WT) and air (AT) during summer. We show that differences in local climate (AT, wind speed, humidity, irradiance), land cover (forest, urban, agriculture), geomorphology (elevation, area/depth ratio), and water transparency explain >30% of the difference in rate of lake heating compared to that of the atmosphere. Importantly, the rate of lake heating slows as air warms ($P < 0.001$). Clear, cold, and deep lakes, especially at high elevation and in undisturbed catchments, are particularly responsive to changes in atmospheric temperature. We suggest that rates of surface water warming may decline relative to the atmosphere in a warmer future, particularly in sites already experiencing terrestrial development or eutrophication.

Climate change has significantly altered lakes worldwide, and is expected to exacerbate current threats to ecosystems and humanity[1,2]. Lakes are central to hydrological, biogeochemical, and ecological processes, thus knowledge on their responsiveness to climate change is essential to their management and maintenance of ecosystem services[3]. In particular, recent research has focused on patterns and apparent sensitivity of lakes to atmospheric warming, due to the critical role of lake water temperature (WT) in regulating ecosystem processes, such as organismal growth, biogeochemical cycles, and food-web interactions[2]. Due to the high specific heat of water, lake temperatures are often buffered against high frequency meteorological variation, and instead integrate longer-term (monthly-to-annual) changes in energy fluxes associated with climatic variability[4]. As a result, the characteristics of ice cover, stratification, surface temperature, evaporation, and water level have all changed notably in recent decades in response to climate warming[2]. There is also growing concern that elevated atmospheric temperatures (AT) are enhancing

symptoms of eutrophication, such as the frequency, magnitude, and geographic extent of cyanobacterial blooms[5–7]. Therefore, understanding the response of lake water temperature to climate warming is critical for predicting biotic change and anticipating the repercussions of climatic variability on lakes and associated ecosystems[8].

Recent studies have documented multi-decadal trends in lake water temperature, suggesting widespread increases in lake surface WT in response to atmospheric warming[8–12]. For example, Jane et al.[9] indicated that lake surface WT in temperate zone increased 0.39 °C per decade from 1980 to 2017, whereas AT increased at 0.30 °C per decade over the same period. In addition to differences in rates of air and water temperatures change, individual lakes exhibited a wide range in rate and magnitude of surface WT change, even including whole-lake cooling despite atmospheric warming in some instances[13]. These findings emphasize the importance of accounting for factors that control heat budgets of basins, rather than assuming that WT responds uniformly to increases in AT.

[1]Nanjing Institute of Geography and Limnology, Chinese Academy of Sciences, 73 East Beijing Road, Nanjing 210008, China. [2]School of Geography, Nanjing Normal University, No.1 Wenyuan Road, Nanjing 210023, China. [3]Limnology Laboratory, University of Regina, Regina, SK S4S 0A2, Canada. [4]Institute for Environmental Change and Society, University of Regina, Regina, SK S4S 0A2, Canada. [5]Department of Biological Sciences, Rensselaer Polytechnic Institute, Troy, NY 12180, USA. ✉e-mail: Peter.Leavitt@uregina.ca; kshi@niglas.ac.cn; qinbq@niglas.ac.cn

Heterogeneity in the rate of lake warming may prevent simple statements about lake WT trends[13] and underscores the importance of considering possible controls warming, including climate, watershed characteristics, lake geomorphometry, and in situ trophic conditions. Generally, climatic features (e.g., irradiance, humidity, wind speed) are expected to be the predominant factors regulating differences in the rates of lake and atmospheric warming[3,14], while parameters controlling the redistribution of heat within the lake have secondary effects on lake warming[10]. Indeed, variations in lake geomorphology (e.g., depth, water residence time, elevation)[15–17], watershed characteristics (e.g., land use)[18,19], and trophic status (e.g., water clarity)[17,20,21] can modulate climate effects on individual lakes by affecting how energy is distributed with depth. For example, Woolway et al.[16] suggests that cold and deep lakes respond more rapidly to variation in AT, while others have found shallow lakes are more sensitive to air warming[15,22]. Rose et al.[21] indicates that lake WT response to AT changes varied among sites in part to differences in water clarity and lake depth. This variation in the responsiveness of WT to atmospheric conditions highlights the heterogeneous and complex responses of lakes to climate and other stressors and makes it difficult to predict the risk of ecosystem damage due to climate change. Further, to date, most mechanistic inferences have been drawn from numerical simulation experiments and still require validation using extensive lake observations. As lake ecosystems are already under serious threat from numerous human-induced stressors (e.g., eutrophication[7], deoxygenation[9]), it is vital to understand where and how global climate change will augment the effects of existing stressors on these important ecosystems[22,23], and to implement this knowledge for future management and conservation strategies[24].

Here variation in the responsiveness of lake surface and deep WT to atmospheric warming during summer was analyzed by comparing long-term (1979–2017, 24.5 ± 6.7 years) estimates of WT profiles and trophic state from 345 north temperate lakes and reservoirs (Fig. 1) with measures of local climate, lake geomorphometry, and watershed characteristics. We hypothesized that the responsiveness of lake WT to atmospheric warming, and the consequent risk of lentic ecosystems to fundamental changes, is not the same for all lakes, and that clear, cold, and deep basins are more sensitive to AT change. This study aims to improve our understanding of the controls of spatial and temporal variation in lake response to atmospheric warming, help decision-makers prepare for future risks, and develop targeted management strategies.

## Results

### Long-term variations of air and lake water temperatures

According to observed lake temperature profiles and lake stratification regimes (stratified, unstratified), lake warming was estimated for epilimnetic (surface) and hypolimnetic (deep) waters. In this study, AT, epilimnetic water temperatures (ET), and hypolimnetic water temperature (HT) in summer (hereafter from July 15 to August 31 in the Northern Hemisphere) were 20.6 ± 2.7 °C, 22.0 ± 3.0 °C, and 9.6 ± 3.0 °C, respectively (Fig. 2a). Many northern lakes exhibited warming trends (as Sen's slope) in both AT (91.0%) and ET (81.7%) during summer, although deep waters changed less consistently, with more than half of sites (58.5%) cooling over the analytical period (Fig. 2b). Epilimnetic temperature generally warmed more rapidly (+0.44 ± 0.57 °C per decade) than did summer AT (+0.36 ± 0.33 °C per decade), whereas HT often declined (−0.12 ± 0.47 °C per decade, Fig. 2b).

For individual lakes, the multi-decadal trends between AT and WT often diverged or even showed opposite trends (Fig. 2c–f). For example, epilimnetic temperature trends (ETT) in 62 lakes (18.0% of sites) and hypolimnetic temperature trends (HTT) in 132 lakes (57.6%) were opposite to the air temperature trends (ATT, Fig. 2d, e), while 55.5% of ETT were opposite to HTT (Fig. 2f). Calculated differences in AT and WT trends varied with lake zone, including the trend differences between ETT and ATT (ETT–ATT, +0.08 ± 0.52 °C per decade), HTT and ATT (HTT–ATT, −0.47 ± 0.53 °C per decade), and ETT and HTT (ETT–HTT, +0.59 ± 0.62 °C per decade) (Fig. 2c). In general, lake surface and deep WT responded differently to atmospheric warming, and there was a wide range in rate of WT change in individual lakes (Fig. 2d–f).

### Controls of the responsiveness of lake water temperature to air temperature

In this study, the responsiveness of WT to changes in AT during summer was evaluated by calculating the difference between trends in WT and AT for each lake. Analysis with pairwise correlations showed that ETT–ATT was correlated positively with lake volume, forest cover, wetland extent, total summer precipitation (TSP), summer longwave radiation (LR), regional summer latent heat flux (*LE*), winter AT (WiAT), and humidity ($P < 0.05$), and negatively with the degree of anthropogenic development, agriculture, grassland area, summer wind speed (WS), regional summer sensible heat flux (*H*), and summer AT (SuAT) ($P < 0.05$, Fig. 3a). Similarly, HTT–ATT was correlated positively with lake area, maximum depth (max depth), ratio of area to depth (area/depth ratio), volume, watershed area (Wshd), forest cover, and WiAT, and negatively with landscape development, SuAT, spring AT (SpAT), and Fall AT (FaAT) ($P < 0.05$, Fig. 3a).

Random forest analysis was used to determine which variables were most important in explaining temperature trend differences

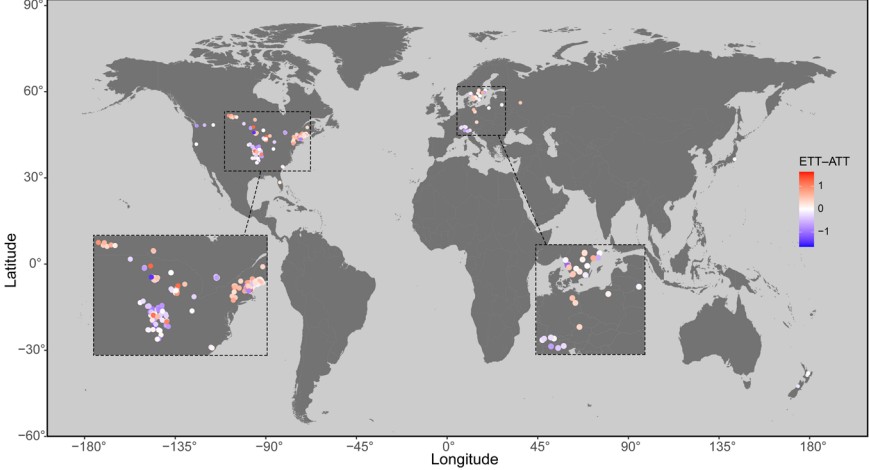

**Fig. 1 | Distribution of 345 lakes used in this study.** Color gradient of circles indicate the differences between lake epilimnetic temperature trend (ETT) and air temperature trend (ATT), as ETT–ATT. Source data are provided as a Source Data file.

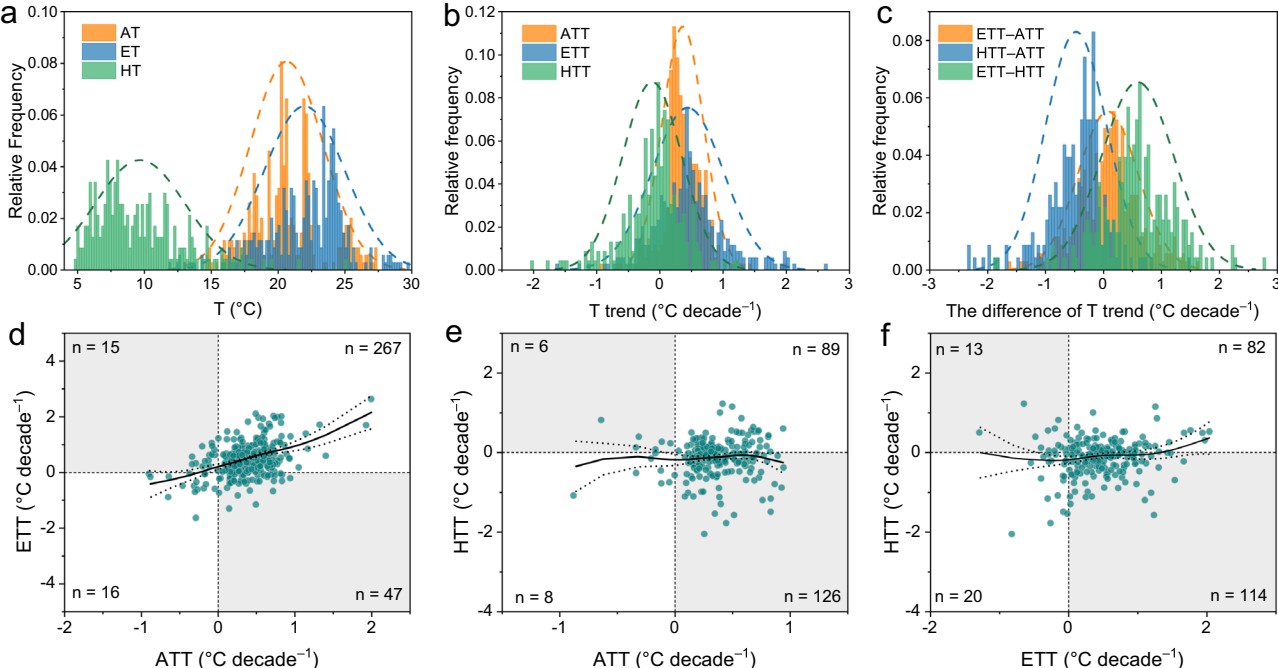

**Fig. 2 | Long-term variations of air and lake water temperatures. a** Relative frequency of air temperature (AT), epilimnetic temperature (ET), and hypolimnetic temperature (HT). **b** Synchronous distribution of air temperature trend (ATT), epilimnetic temperature trend (ETT), and hypolimnetic temperature trend (HTT). **c** Density plots of differences among trends in air, epilimnetic, and hypolimnetic temperatures. ETT−ATT, the difference between ETT and ATT in summer; HTT−ATT, the difference between HTT and ATT in summer; ETT−HTT, the difference between ETT and HTT in summer. **d**−**f** the relationships among ATT, ETT, and HTT. Source data are provided as a Source Data file.

between air and water. More than 30% of the variation in the relative rates of air and lake temperature change was correlated to local differences in geomorphic, watershed, climatic, and trophic characteristics (Fig. 3b, c). According to the random forest analysis, the set of predictors used in this study explained 30.3% of ETT−ATT and 31.1% of HTT−ATT, respectively ($P < 0.001$, Fig. 3b, c). Specifically, WS, humidity, SuAT, WiAT, elevation, forest cover, summer shortwave radiation (SR), urban development, wetland, the lake area, and grass were the important factors explaining the differences between ETT to ATT ($P < 0.05$, Fig. 3b), while variations in HTT−ATT were significantly explained by changes in SuAT, forest cover, LR, urban development, shrubland, and water transparency (Secchi depth) ($P < 0.05$, Fig. 3c). Moreover, a variance partitioning analysis showed that variations in ETT−ATT were primarily explained by climate (21.6%), lake geomorphology (2.7%), land use (0.6%), and the combined effects of land use and climate (5.3%) rather than to water transparency ($P < 0.05$, Fig. 3d). In contrast, differences in HTT−ATT were predominantly related to climate (21.7%), land use (5.6%), and water transparency (1.5%) rather than to lake geomorphology ($P < 0.05$, Fig. 3d).

Analysis with generalized additive models (GAMs) revealed that values of ETT−ATT declined significantly with AT warming during summer (Spearman's r = −0.267, $P < 0.001$, Fig. 4). On average, trend differences between ETT and ATT tended to be negative in warm regions (Fig. 4), indicated that surface WT was less responsive to changes in AT in regions with warmer climates.

### Effects of trophic status on lake responsiveness to atmospheric warming

Following Organization for Economic Co-operation and Development (OECD) guidelines[25], Secchi depth values were used to categorize lakes according to trophic status. Transparent oligotrophic lakes exhibited greater responsiveness to AT change than did productive turbid sites (Fig. 5). The difference between trends in WT and AT were correlated positively with Secchi transparency ($P < 0.001$, Fig. 5), which was significantly lower in eutrophic and hypereutrophic lakes compared with oligotrophic and mesotrophic sites ($P < 0.05$, Fig. 5). For example, mean ETT−ATT values declined progressively with lake trophic status from oligotrophic ($0.20 \pm 0.36$ °C per decade) to mesotrophic ($0.19 \pm 0.59$ °C per decade), eutrophic ($0.001 \pm 0.53$ °C per decade), and hypereutrophic ($-0.17 \pm 0.49$ °C per decade) basins (Fig. 5a), indicating that surface water warmed more slowly in turbid (hypereutrophic) lakes than did the local atmosphere ($P < 0.05$, Fig. 5a). Similarly, HTT−ATT values increased significantly with Secchi depth values ($P < 0.05$, Fig. 5b), and deep water warmed slowly compared with AT, especially in more productive sites ($P < 0.05$, Fig. 5b). Analysis with a subset of lakes using nutrient content (as TP) or phytoplankton abundance (as Chl $a$) confirmed that more productive lakes exhibited lower sensitivity to rising air temperatures than did unproductive systems (Supplementary Fig. 1).

## Discussion

Consistent with previous studies of widespread environmental change[9,13], summer air temperature in this study increased by $0.36 \pm 0.33$ °C per decade, shortwave radiation increased by $1.70 \pm 3.4$ W m$^{-2}$ per decade, while wind speed and precipitation declined by $0.04 \pm 0.05$ m s$^{-1}$ per decade and $1.81 \pm 24.1$ mm per decade, respectively (Supplementary Table 1). Because many stratified lakes exhibited ET increasing more rapidly than local AT, while HT frequently showed a cooling trend[10,23], our results indicate that stratification strength (buoyancy frequency) also increased ($0.00019 \pm 0.0012$ s$^{-2}$ per decade), while the depth of stratification declined ($-0.15 \pm 0.65$ m per decade) due to atmospheric warming (Supplementary Fig. 2)[9,26]. Climate warming, decreased wind speed, and increasing solar radiation (Supplementary Table 1) all interact to cause lakes to exhibit earlier and more prolonged thermal stratification[26,27], decreased epilimnion thickness[28,29], dampened water mixing, and reduced thermal diffusivity in the thermocline during

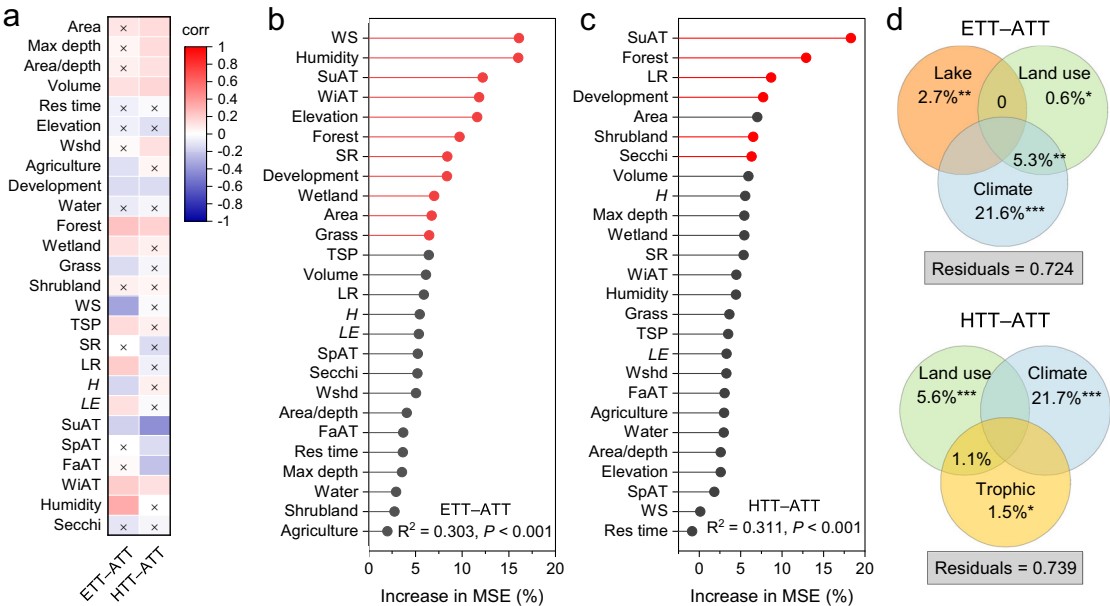

**Fig. 3 | Controls of the responsiveness of lake water temperature to air temperature. a** Pairwise correlations between differences in trends of epilimnetic water temperature (ETT) and atmosphere temperature (ATT) (i.e., ETT-ATT) or hypolimnetic water temperature (HTT) and atmosphere temperature (i.e., HTT-ATT) and key environmental parameters. Environmental predictors include lake geomorphometry (area, maximum depth [max depth], area/depth ratio [area/depth], volume, water residence time [res time], elevation, and watershed area [wshd]), land use (agriculture, development, water, forest, wetland, grass, shrubland), climate (trends in wind speed [WS], total summer precipitation [TSP], humidity, shortwave radiation [SR], longwave radiation [LR], regional sensible heat flux (*H*), and regional latent heat flux (*LE*) during summer as well as summer air temperatures [SuAT], spring air temperature [SpAT], fall air temperature [FaAT], and winter air temperature [WiAT]), and trophic state (trend in Secchi depth [Secchi]) examined by Spearman's correlation coefficient. The color gradient indicates the correlation coefficients (corr) and the squares with a cross indicate non-significant correlations (*P* > 0.05). **b, c** Importance of lake geomorphology, land use, climate, and trophic variables in explaining the ETT−ATT and HTT−ATT explored using random forest analysis. The explanatory power of response variables was estimated as the mean squared error (MSE). Statistical significance indicated by red plot. **d** Variation partitioning analysis of the relative contributions of lake geomorphology (lake), land use, climate, and trophic variables to the response of lake water temperatures to air temperature. The values < 0 were not shown. Statistical significance indicated by *P* < 0.05, **P* < 0.001, and ***P* < 0.001 (ANOVA). Source data are provided as a Source Data file.

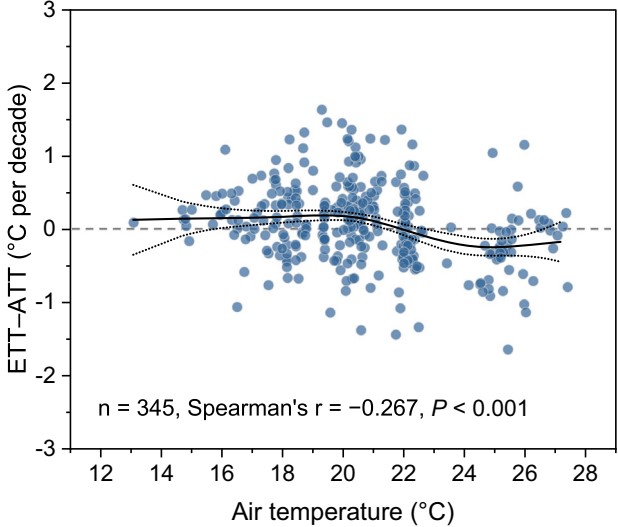

**Fig. 4 | Differences in trends of epilimnetic water temperature and atmospheric temperature change as a function of air temperature.** Lakes become less responsive to air temperature change as the atmosphere warms. The black line is a generalized additive model fit to the data points, whereas the fine dotted line represents the pointwise 95% credible interval of the fitted values. Pairwise correlations between epilimnetic temperature trend (ETT) and air temperature trend (ATT) (i.e., ETT−ATT) and air temperature were examined by Spearman's correlation coefficient. Source data are provided as a Source Data file.

periods of lake stratification[22]. Accordingly, reduced heat diffusion and water-column mixing caused surface waters to warm disproportionately, while deep waters showed limited warming, with cooling even observed in many stratified lakes[10,22,23,30]. Consistent with this mechanism, WT more closely tracked changes in AT in shallow or nonstratified lakes, while surface and deep waters in stratified lakes exhibited a greater range of trends (Supplementary Table 2).

The impact of climate change on lake surface WT has been extensively studied and discussed[13,31,32]. Instead, this study focused on the relative rates of heating of water and air to describe the heterogeneity in lake warming, and evaluate the importance of interactions among climate, watershed, and geomorphic factors in regulating lake temperature responsiveness to atmospheric warming. Here, random forest analysis and variation partitioning analysis showed that climate and watershed characteristics were the two most important measured factors explaining differences in trends between WT and AT (Fig. 3). Like some numerical climatic models[3,14], we find that AT is a key driver for changes in lake surface temperatures on a global scale (Fig. 3). However, we also note that AT was only one of a series of climatically-related parameters that predicted differences in trends between air and water temperatures (Fig. 3b, c), suggesting that more comprehensive analyses will be needed in the future to predict lake warming. As well, we note that lake-specific geomorphic properties (e.g., lake depth, elevation, area/depth ratio) also affected differences in air and water warming trends, both directly and through interactions with climate drivers (Fig. 3)[10,15,19]. For example, lake surface area and depth (and their ratio) affect the strength of stratification and can result in a net decrease in the whole lake average temperature[17]. Finally, we recorded that the factors

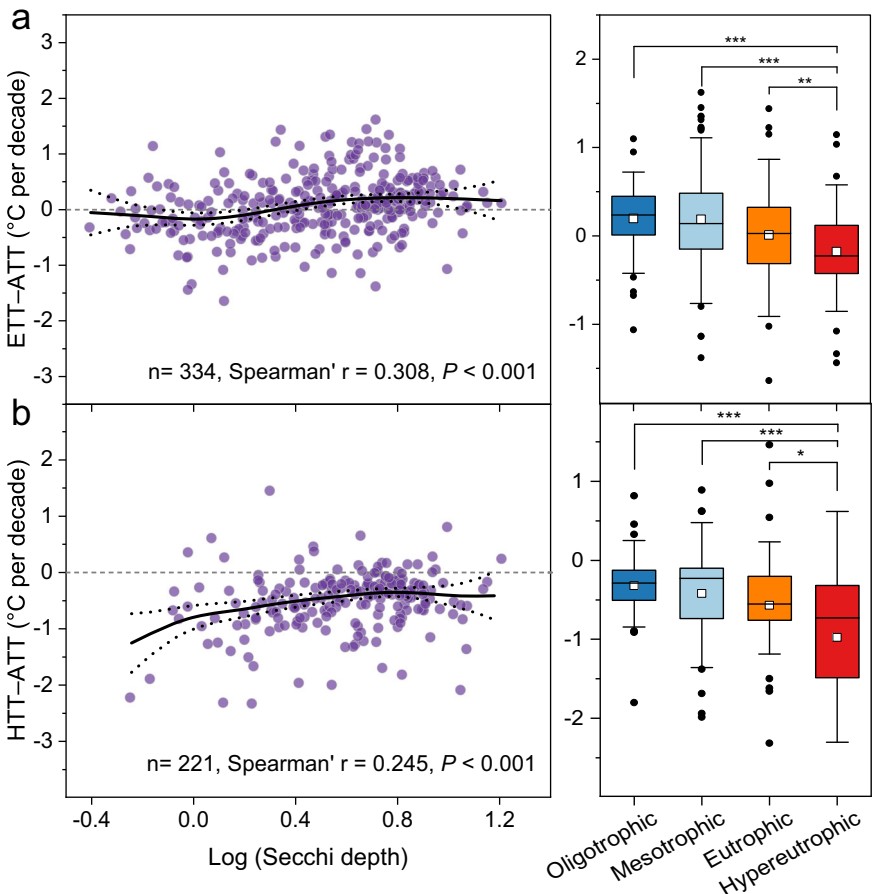

**Fig. 5 | Distribution of trend differences between water temperatures to air temperature as a function of water transparency. a** The relationships between epilimnetic water temperature trends (ETT) and air temperature trends (ATT) (i.e., ETT–ATT) and Secchi depth. **b** Variations in hypolimnetic water temperature trends (HTT) and air temperature trends (i.e., HTT-ATT) as a function of water transparency. The black line is a generalized additive model regression, and the dotted line represents pointwise 95% confidence interval of the fitted values. Pairwise correlations between ETT-ATT and HTT-ATT and Secchi depth were examined by Spearman's correlation coefficient. Secchi depth data were log$_{10}$-transformed. According to the Organization for Economic Co-operation and Development (OECD), lakes were classified as oligotrophic (Secchi > 6 m), mesotrophic (3 m <Secchi ≤ 6 m), eutrophic (1.5 m <Secchi ≤ 3 m), or hypereutrophic (Secchi ≤ 1.5 m). Statistical significance indicated by $^{*}P < 0.05$, $^{**}P < 0.001$, and $^{***}P < 0.001$ (ANOVA). Source data are provided as a Source Data file.

influencing long-term temperature changes in the hypolimnion appear to be distinct from those driving epilimnetic warming, possibly because deeper waters are isolated from the main avenue of energy exchange, the air-water boundary layer[10].

Over the past decades, land use and land cover has been widely recognized as a critical factor mediating socioeconomic, political, and cultural behaviors and global climate change[1]. Human modification of the land surface affects both regional and global climate processes by changing the fluxes of mass and energy between lake ecosystems and the atmosphere[33,34]. Indeed, catchments act as site-specific filters of climate and human effects by altering terrestrial subsidies to lakes[19,33,35]. For example, when land cover is converted to agriculture, sensible heat flux decreases, while latent heat flux exhibits little change[36,37], resulting in warmer regional surface temperatures that can affect local lakes (Supplementary Fig. 3). Conversely, increased forest growth can decrease surface wind speeds and increase aquatic concentrations light-absorbing dissolved organic matter while also intensifying thermal stratification, thereby modifying the effects of atmospheric warming on lake thermal regimes[18]. Although not explicitly addressed in this study, transformation of forest, grassland and wetland habitats to an urbanized environment also likely affects how lakes warm (Fig. 3), both due to heat-island effects and influences on wind speed and direction[38]. However, despite general congruence between our analysis of widely-distributed temperate lakes and findings from numerical models[8,12,27] and site-specific studies[18], we note that further research is required to refine our understanding of the mechanisms by which climate and land-use factors interact to warm lakes.

Cultural eutrophication of surface waters has been an international concern for over 75 years due to its ecological and economic consequences, including harmful algal blooms. More recently, attention has focused on the role that climate change may play in regulating lake production, community composition, and biogeochemistry[7,39]. In particular, atmospheric warming has been linked to a rise in toxin-producing cyanobacterial blooms in freshwater ecosystems worldwide[5–7]. Accordingly, an improved understanding of the responsiveness of lake WT to climate change will help guide adaptation strategies[30]. This study suggests that clear and unproductive lakes may be more sensitive than turbid productive lakes to atmospheric warming (Fig. 5 and Supplementary Fig. 1)[40]. Increased light penetration is associated with increased heating of deep waters and elevated mean water-column temperatures[40], whereas productive waters often exhibit reduced sunlight penetration, mixing depths, and hypolimnetic temperatures, even though trends may vary somewhat among basins[3,17,20,21,30]. Together, these patterns indicate that the interactive effects of human activities (e.g., land use and lake trophic state) are important in shaping the response of lake water temperature to climate change.

Forecasts of lake response to future atmospheric heating often focus on substantial changes associated with changes in physical properties (ice cover, stratification regime) rather than progressive changes in lake responsiveness to increase AT. We find that the responsiveness of surface WT to atmospheric warming appeared greatest when lakes were cool or located in a colder climate such as occurs at high latitude or elevation[15], and declines as the atmosphere warms (Fig. 4). In general, this relationship reflects the observation that ETT, but ATT, declines with increased AT (Supplementary Fig. 4), consistent with the expected effects of increased evaporation and latent heat transfer to the atmosphere at higher temperatures[41]. However, resolution of the precise mechanism(s) leading to a progressive decline in lake responsiveness to atmospheric warming will likely require comprehensive energy budgets to better quantify how the thermal regimes and heat exchange dynamics of lakes vary systematically with AT and other factors.

Understanding how lake WT respond to climate change is important to predict how lake functions may change in the future[2]. Here, responses of lake WT to atmospheric warming were determined to be heterogeneous due to differences in the lake physical and chemical features, watershed characteristics, and local climatic conditions. These results suggest that the responsiveness of lake WT to climatic variations, and consequently the risk of water quality issues, is heterogeneous and that a "one-size fits-all" approach is not appropriate to understand and manage the risks of climate warming[23,42]. Instead, we conclude that it is important to account for differential lake responsiveness to climate warming when developing adaptation and mitigation strategies. Clear, cold, and deep lakes, especially those situated at high elevations and in areas of natural land use, exhibited the highest responsiveness to atmospheric warming, therefore, may be at the greatest risk to experience major ecosystem changes associated with warming[31,40,43]. Similarly, as anthropogenic eutrophication of surface waters continues to increase globally[44], we anticipate that the magnitude of lake responsiveness to atmospheric warming may decline, necessitating an evolution in strategy of lake management in response to climate change. Better understanding of potential lake sensitivity to climate warming may help decision-makers identify sensitive ecosystems, improve our ability to forecast the responses of lake ecosystems to future climate changes, and better prepare for future climate risks (e.g., fish kills, anoxia, harmful algal blooms).

## Methods
### Data set
This study uses a large data set incorporating long-term records of water temperature (WT) profiles, local climatic variables, and lake trophic state, as well as a database of lake geomorphic and watershed characteristics collected by academic, government, and not-for-profit sources[45,46]. Geomorphic characteristics for each lake were sourced from the HydroLAKES project[47] and included surface area, maximum depth, area/depth ratio, volume, water residence time, elevation, and watershed area. Water temperature profiles were derived from in situ measurements and had at least one profile sampled annually during the ice-free period[40]. Meteorological variables derived using the ERA-5 reanalysis from the European Centre for Medium-Range Weather Forecasts[48] included various air temperatures (AT) including spring air temperature (SpAT), summer air temperature (SuAT), fall air temperature (FaAT), winter air temperature (WiAT), as well as summer wind speed (WS), humidity, summer short-wave radiation (SR), summer long-wave radiation (LR), total summer precipitation (TSP), regional summer latent heat flux (LE), and regional summer sensible heat flux (H). Climate data were sourced from locations nearest to each lake. The composition of land use within each lake's watershed was derived from the US national land cover database for most North American sites[46], while Landsat images with a spatial resolution of 30 m

were used to obtain land use data for basins outside the USA[49]. The percentage of each land use category for each watershed was characterized as; agriculture, developed, water, forest, wetlands, grass, and shrubland[46]. In this study, lake trophic state was estimated from measurements of Secchi values from multiple data sources[46].

Summer WT is especially important from a lake ecosystem perspective and was the focus of this study. Based on the interval of stable summer stratification, summer period was defined as the period from July 15 to August 31 for lakes situated in the Northern Hemisphere, whereas the few southern hemispheric locations used the interval January 15 to February 28. Selected lakes had at least 15 years of data between 1979 and 2017. For quality control, metadata for each lake was gap-matched for each variable across data sets. Overall, 345 diverse temperate lakes were available in this study (Supplementary Table 3; Supplementary Data file), mostly located in the Northern Hemisphere (Fig. 1).

### Lake stratification
Lake stratification was calculated from observed lake temperature profiles. If the vertical range in temperature was <1 °C, the water column was considered to be unstratified[45]. If more than 10% of profiles were considered unstratified, the lake was considered not to have a hypolimnion[45]. Epilimnion was defined as all depths less than or equal to the uppermost metalimnion depth, and hypolimnion as all depths deeper than the deepest metalimnion depth. In this database, 229 lakes exhibited stratification, whereas 116 lakes exhibited no stratification during summer (Supplementary data). For lake WT, we calculated the mean of all parameters recorded for the epilimnion and hypolimnion.

### Lake trophic state
The availability of estimates of trophic state varied with parameter and depth. Water transparency (as Secchi depth, m) was more widely available (334 lakes after removing the sites where the transparency reached the bottom of the lake) and was used to categorize lake into four major trophic status, following the OECD[25]; oligotrophic (Secchi > 6 m), mesotrophic (3 m <Secchi ≤ 6 m), eutrophic (1.5 m <Secchi ≤ 3 m), and hypereutrophic (Secchi ≤ 1.5 m). Thus defined, there were 79, 111, 77, and 67 lakes classified as oligotrophic, mesotrophic, eutrophic, and hypereutrophic, respectively (Supplementary data).

### Long-term trend calculations
To obtain trends for each variable of individual lakes, the annual mean values of climatic variables and all of the epilimnetic and hypolimnetic values were calculated for each lake. Here, Sen's slope, a commonly-used metric for trend analysis of long-term series, was calculated and used to estimate trends in WT (epilimnetic and hypolimnetic), climatic factors (SpAT, SuAT, FaAT, WiAT, WS, humidity, SR, LR, TSP, H, and LE), and trophic state (Secchi) (Supplementary data). Sen's slopes and significance (alpha = 0.05) were calculated in R 4.0.4 using the trend package[50].

### Trend differences between water and air temperatures
In this study, the responsiveness of WT to change in AT during summer was evaluated by calculating the difference between trends in WT and AT for each lake. Specifically, the difference between epilimnetic temperature trend (ETT) and air temperature trend (ATT) in summer (ETT−ATT) were used to indicate the responsiveness of epilimnetic temperature (ET) to AT. Similarly, the responsiveness of hypolimnetic temperature (HT) to AT in summer was calculated as the difference between the hypolimnetic temperature trend (HTT) and ATT, as HTT−ATT. Here, negative values between ETT and ATT indicate that lake water is warming more slowly or cooling faster than the atmosphere.

## Regression tree analysis of the trend differences between water and air temperatures

Random forest analysis was used to determine which variables were most important in explaining the responsiveness of WT to changes in AT[51]. Predictors included geomorphic (lake area, maximum depth, area/depth ratio, volume, water residence time, elevation, watershed area), and watershed characteristics (agriculture, developed, water, forest, wetlands, grass, shrubland), as well as the trends in water transparency (Secchi) and climatic variables (WS, humidity, SR, LR, TSP, $H$, $LE$, SpAT, SuAT, FaAT, WiAT). The order of importance was determined by the frequency of variables and their relative position in individual trees across the entire forest. The explanatory power of response variables was estimated as the mean squared error (MSE). Subsequently, preliminary models of interannual variation in WT were used to assess whether the significant variables offered reasonable predictions of WT responsiveness to changes in AT. Random forest analysis was conducted using the *randomForest* package in R 4.0.4[52]. In addition, the *A3* R package was used to assess the significance of the models and cross-validated $R^2$ values with 5000 permutations of the response variables[53]. For each analysis, only lakes with no missing values for any predictor variables were used.

## Variation partitioning analysis

To assess the relative effects of lake geomorphic, watershed, trophic, and climatic variables on the relationship between water temperature trends and ATT, a variation partitioning analysis was performed using 'varpart' function of the *vegan* package in R 4.0.4[54]. Differences in trends of WT and AT were used as the response variable to four sets of explanatory variables: lake geomorphology (area, maximum depth, area/depth ratio, volume, water residence time, elevation, watershed area); watershed characteristics (agriculture, developed, water, forest, wetlands, grass, shrubland); climatic features (WS, humidity, SR, LR, $H$, $LE$, TSP, SpAT, SuAT, FaAT, WiAT), and; trophic state (Secchi). Multiple regression using both forward and backward selection was used to reduce collinearity among predictors in each of the explanatory sets[55]. For ETT–ATT, eight variables (WS, agriculture, SuAT, humidity, elevation, SpAT, SR, area/depth ratio) were selected, while four variables (SuAT, forest, Secchi, WiAT) were retained to explain HTT–ATT. Variation partitioning was performed to evaluate the direct and interactive correlations between the climatic, watershed, geomorphic, and trophic predictors of WT responsiveness to changes in AT. All partitioning fractions of variation were significant in an analysis of variance (ANOVA) permutation test using the *vegan* package in R 4.0.4[54].

## Statistical analysis

The distributions of ETT–ATT and HTT–ATT with air temperature and water transparency were estimated using generalized additive model (GAM) in the *gam* package[55]. Statistical relationships among geomorphic, watershed, climate, and trophic conditions were examined with a one-way analysis of variance (ANOVA) using Tukey's Honestly Significant Difference test. Correlations between WT trends and lake geomorphic, watershed, climatic, and trophic variables were explored with Spearman's correlation coefficient using the *stats* package. All analyses were performed in R 4.0.4[56]. The level of significance used for all tests was $P < 0.05$.

## Data availability

The underlying raw data used for the analysis in this study are openly accessible online from https://doi.org/10.6073/pasta/ac8b05bb0da19032b3df3efc21f83874[45] and https://doi.org/10.6073/pasta/312f45d8d2ceaecf0c02e791f5fd9a63[46]. The data of sen's slope are available as Supplementary data. Source data for the figures are provided in the Source Data file. Source data are provided with this paper.

## Code availability

The source R code used in this study are publicly available at https://github.com/Laker-NIGLAS/Source_code[57].

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

## Acknowledgements

This work was supported by the National Natural Science Foundation of China (grants 42220104010 to B.Q., 42177058 to J.Z., U22A20561 and 41922005 to K.S., and 42007160 to Y.Z.), the National Key Research and Development Grant of China (2022YFC3204101) to J.Z and the NIGLAS foundation (E1SL002) to K.S., the Canada Research Chair Program to P.R.L., and US National Science Foundation (grants 1754265 and 2048031) to K.C.R.

## Author contributions

J.Z. proposed the idea, designed the research, conducted data analysis, created figures, and wrote the draft of the paper; P.R.L. refined concepts and contributed to research design, data compilation, analysis and manuscript preparation; K.C.R. contributed to data compilation and manuscript preparation; X.W. contributed to data compilation and analysis; Y.Z. helped prepare the manuscript; K.S. refined concepts and contributed to data compilation, analysis and manuscript preparation. B.Q. refined concepts and helped prepare the manuscript. B.Q., P.R.L. and K.S. supervised the project.

## Competing interests

The authors declare no competing interest.
