## [Peer Review File · Nature Communications]

REVIEWER COMMENTS

Reviewer #1 (Remarks to the Author):

General comments

This is a valuable paper, interesting, novel and generally well written and presented. I have some concerns about terminology and lack of quantification of statements, as well as other points below. With due adjustment relative to the comments below, I will be happy to recommend publication.

Specific comments

Abstract -- many of the points could be better quantified in the abstract, such as the magnitude of the sensitivities (I assume in K/K) being discussed, what a "significant decrease" is quantitatively, etc. Also, I find the "threshold" language on sensitivity is not really justified. Such language suggests a step change in sensitivity, which is not credible from the data. Even though you have imposed a two-fit model on it with an inflection point, use of "threshold" to describe that inflection point is misleading, both because there is no step change in sensitivity, and because the step change in slope of sensitivity is a feature of your choice of fit rather than a feature of the data.

Line 43 -- What is long term? Quantify. Lakes integrate over the weather on a range of timescales that is very variable between lakes and can be just days, ie, not really "long term".

Line 44 "slowly developing" -- quantify, as above. Ice can form overnight. Is that slow? Suggest reword to be clear.

Line 53 -- O'Reilly is a good reference, but it is pretty old (trends before 14 years ago -- a lot has happened since then). There are annual updates on trends in lake temperature and air temperature in the annual BAMS state of the climate reports that could be a source of more contemporary statements, and probably other papers.

Line 57 - Grammar correction needed for "apparently cooling".

line 58 - There is no mystery that not all lakes follow air temperature, because insolation, windiness, etc are well known to be important. So a better wording is "importance of accounting for the other factors" rather than "understanding".

Line 62 - I agree with these additional factors discussed, but the other meteorological factors are omitted. There are trends in insolation, cloudiness (and therefore radiation budget) and wind in some areas, and these modify lake temperature - air temperature relationships too. In fact -- you point this out later, but I think it should be noted together here.

Line 78 --What is the definition of "lake temperatures" here? Is it annual average, summer, ...? What depth(s)? This is in the methods section later, but it needs to be stated here for those reading from start to finish in the order presented.

Line 82 -- Here you mention irradiance -- I think it should be mentioned a little earlier as per previous comment

Line 92 -- figure 2 is referenced before figure 1 which is non-standard.

Figure 2 a and b -- there are 347 lakes, which can't give these smooth distributions. The visual impression can depend on the fit made to give these smooth distributions, and I would prefer to see histograms.

Figure 2d - I would be more interested in the HTT vs ETT plot than HTT vs ATT, or maybe as well as?

Line 104 -- again, a sentence framing the HTT changes vs ETT is just as valid to include. The variability of HTT vs ETT seems interesting to me.

Line 107 -- The difference in trend is defined as "sensitivity". I wouldn't like this word for this quantity to become established as I think it is confusing. A sensitivity is universally a differential dx/dy , not $dy(dx/dy - 1)$ which is your definition. You are calculating "trend differences", and I don't

see a problem calling them trend differences. Always better to call something what it is rather than what it is not.

For an example of why "sensitivity" is a bad name, consider a case, which could happen in your dataset, of a lake with ETT -0.1 K/dec and ATT +0.1 K/dec. In what sense does "a sensitivity of -0.2 K/dec" make sense: clearly the ETT in such a case is not driven by the ATT and is responding to other factors, rather than responding to ATT with a "negative sensitivity".

Therefore please replace "sensitivity" with the more accurate "trend difference" throughout.

Line 109 -- "Analysis with random forest showed " is not very clear, please amend.

Line 116 -- VPA has not been expanded as an acronym.

Fig 3 caption -- need to expand acronym MSE in the caption as this is read before the methods section.

Line 124 -- seems to me to be more accurately phrased as "trend differences on average tended to be negative for $ATT > X$ deg C, and close to zero on average below that temperature". The reason is that your analysis shows many other factors are important and these may be confounding the effect: the warmer lakes warm slightly less fast than air temperature, but potentially because of your other influences and not because of a lower actual (local) sensitivity (in the dx/dy sense) to AT.

Line 138 -- this is the first time that it is mentioned that you are looking at "summer" air temperature -- needs to be clear before this. Including what "summer" means.

Line 138 -- the numbers are averages and there is variability. I think we need to have the fact these are means across 347 locations explicitly stated, and suggest to quote the variability (SD) as well as the mean values ($X \pm Y$).

Line 168 -- the argument here seems to be that the land-use effect is mediated by altering the local air temperature -- but in that case why isn't all of that wrapped up in the SuAT factor, which is then correlated?

Line 176 -- I think this speculation also needs a brief account of why you might think that -- e.g., the warm lakes are in locations of higher water vapour that inhibits the nighttime net radiative cooling of the lake.

Line 189 -- I don't find this very convincing as a causal explanation. You would need to rule out that it arises from the sampling effect you then go on to discuss, I think, in order for it to be worth making this speculation. Can you rule out that it is all to do with confounding?

Line 210 -- You need to quantify the statements so that the size of this alleged reduction in sensitivity is clear. Looking at your plots, if a lake followed the red lines in figure 4, a lake that warmed from 24 degrees (at which is true sensitivity is on average 1 K/K) to 30 degrees would thereafter have a sensitivity of about 0.8 K/K. Another way to look at it is that 3 decades of mean summer T warming of 0.34 K/decade will on average reduce the sensitivity from ~ 1.0 K/K on average to ~ 0.965 K/K. This modest effect (which I don't even necessarily trust because I think you are seeing a confounding effect) should be matched by comparably modest wording of the finding. I find the current wording without quantification to give an exaggerated impression of importance. Please reword with appropriate quantification and appropriate explanation that the "reducing sensitivity" is at most a small effect compared to the lake warming itself.

Figure 4 and similar -- I suggest you need to add the zero line across the plot so we can see differences of the redline from zero more easily.

Reviewer #2 (Remarks to the Author):

Key Results

This paper presents a correlational analysis of thermal sensitivity of temperate lakes to atmospheric warming using information morphology and multi-decadal thermal profiles of 347 lakes. The paper is of interest because of the large data set assembled, which has not been common in the literature. Among the key points noted is that a majority of temperate lakes show increases in summer epilimnetic temperature over several decades, but greater than 50% of the studied lakes showed cooling trends in the hypolimnion. This is not a new finding and reflects complex interactions between onset of stratification, stratification strength, and seasonal weather patterns that have been predicted through modeling (e.g., Stefan et al., 1996), but it is nice to see this finding

confirmed across an extensive data set. The paper also states that “sensitivity of surface WT to atmospheric heating decreased significantly when AT exceeded 20.2°C, suggesting a threshold for strong climate effects on lakes. Specifically, surface WT sensitivity to atmospheric warming may decline in a warmer future...” If true, this would constitute good news as a water temperature around 20 °C approximates a threshold for health of many cold water fish species. Unfortunately, I do not think the data fully supports this conclusion.

Validity

The data analysis is based largely on application of random forest classification (followed by variation partitioning analysis and piecewise regression) to the difference between trends in water temperature and air temperature, which the authors refer to as “sensitivity.” Little information is provided on the calculation of trends except to state that they are based on Sen’s slopes. Sen’s non-parametric slope estimate is usually based on the median of slope estimates across multiple stations; it is apparently used here to represent a slope estimate at a given station over a given time period. There is no discussion of how these slope estimates may be biased by different start and end periods (as, for instance, might be expected by longer-term SST anomalies) for individual lakes.

The validity of the suggested threshold in response at AT of 20.2°C seems questionable. This appears to be obtained from the breakpoint in the piecewise regression in Figure 4(a) (line 502) and is discussed at line 172 of the text. As noted at line 171-177, “effects of atmospheric warming on ET diminished when lakes were in warmer regions (AT > 20.2°C), or were themselves already warm (> 23.8°C). Because air temperature was correlated to trophic status...it is possible that diminished sensitivity to AT reflects arises because eutrophic lakes are often warmer than clear lakes... Resolution of the underlying mechanism will likely require comprehensive energy budgets, although we speculate that warm lakes may be more likely to lose heat to the atmosphere at night in temperate regions...” Figure 4(a) suggests the evidence for the breakpoint is weak while the discussion acknowledges that differences among climatic regions may also have a significant impact on the finding. Thus, I think it is inappropriate to bring the suggested breakpoint forward as a major finding in the Abstract.

In general, the validity of inferences presented in the paper is difficult to ascertain as the analysis is based on correlation or classification and not on causal hypotheses.

Significance

I am pleased that the authors have assembled and analyzed a large dataset of temperate lakes. While the analysis is relatively simple and does not break new ground, surveys across large datasets such as this are uncommon in the literature. I think, however, that the paper could be greatly

improved by comparison to hypotheses of lake response to atmospheric warming obtained from simulation studies.

Data and Methodology

This paper is a summary analysis across a large number of lakes, so I do not expect all data to be provided with this paper or its supplemental information. The authors state that all data are available in references [44] and [45]. Can they affirm that this is true? I think it would be advisable to include summary information (particularly tabular information on Sen's slopes for each of the relevant variables) in the supplemental information for this paper.

Analytical Approach

The analytical approach consists of applying random forest analysis to the difference in trends (i.e., rate of change) in air temperature and water temperature, followed by variation partitioning analysis of groups of explanatory variables and piecewise linear regression. The potential explanatory variables include various measures of climate (e.g., summer air temperature, wind), lake morphology, and watershed characteristics. The analyses appear to be correctly implemented, although I would suggest supplying the R code in supplementary information.

While the relationships of trends in epilimnetic and hypolimnetic summer water temperatures to various explanatory variables is statistically significant, they only explain a small portion of the total variation with an unexplained residual of 80% for epilimnetic water temperature and 74% for hypolimnetic water temperature. Climate inputs have the greatest explanatory power, while, somewhat surprisingly, lake morphological characteristics did not emerge as a significant contributor to hypolimnetic temperature.

The variation analysis is a useful exploratory tool, but can only evaluate information that is supplied to it. The low amount of explained variation may reflect the complexity of interactions between various inputs or the lack of information on key factors. For instance, water clarity may play an important role in the relative temperature trends in surface and bottom waters, but water clarity is not an explanatory variable in the variation analysis (despite Secchi depth being used in other parts of the paper), although it is likely correlated with some of the land use variables. For lake morphology, surface area and depth taken individually are not significant predictors, but it is more likely the ratio of these variables that is a significant influence on mixing regime.

The piecewise linear regression is used to propose a breakpoint in the response of water temperature to air temperature, which seems suspect as noted above. Additional statistical analysis is needed on this point to determine whether the apparent change in response slope at the breakpoint is actually significant.

Suggested Improvements

The paper suggests relationships between trends in atmospheric temperature and lake water temperature in both the epilimnion and hypolimnion of temperate lakes. However, the analysis is primarily correlational in nature and presents (or perhaps confirms) findings that have already been suggested by others. The paper could be greatly improved by reframing the analyses as an evaluation or test of hypotheses that have been proposed in the existing literature.

Many of the responses evaluated here have been investigated previously through model experiments related to climate change. It would be useful to compare and contrast some of these inferences to the results reported here. Some of these findings include the following:

- Lake surface area and depth (and especially their ratio) affect strength of stratification and can result in a net decrease in whole lake average temperature (Stefan et al., 1996; Tanentzap et al., 2008).
- Increased water clarity can warm deeper waters; conversely reduced clarity can reduce the depth of light penetration as well as the mixing depth and reduce hypolimnetic temperatures, although responses in individual lakes are complex (Stefan et al., 1996; Hocking and Straškraba 1999; Rose et al., 2019).
- In addition to effects of lake morphometry, anomalies in the responses of individual lakes to increased atmospheric temperature is modulated by changes in relative humidity and wind speed (Schmid et al., 2014; Woolway et al., 2019).

Clarity and Context

The writing is generally good; however, organization of the paper could be improved. As currently structured, the paper contains two major section heads (Results, and Methods) with no subheads. The Nature style is to put details of Methods after Results; however, it would be helpful to put somewhat more summary information on methods in the Results section if it is presented first. In addition, the long Results section should be divided with subheads to aid the reader.

References

The paper contains a relatively comprehensive survey of relevant literature. The authors should consider also citing the following (see below for complete citations):

- Hocking and Straškraba, 1999
- Rose et al., 2016
- Tanentzap et al., 2008
- Woolway et al., 2019

In addition, Pilla et al. (2020) is mentioned only in passing but likely deserves a bit more discussion.

References Cited in this Review

Hocking GC, Straškraba M. (1999) The effect of light extinction on thermal stratification in reservoirs and lakes. *Int Rev Hydrobiol* 84: 535-556, doi:10.1002/iroh.199900046

O'Reilly, C. M. et al. (2015) Rapid and highly variable warming of lake surface waters around the globe. *Geophys. Res. Lett.* 42(24), 773-781.

Pilla, R. M. et al. 2020. Deeper waters are changing less consistently than surface waters in a global analysis of 102 lakes. *Sci. Rep.* 10(1), 20514.

Rose, K.C., L.A. Winslow, J.S. Read, and G.J.A. Hansen. (2016) Climate-induced warming of lakes can be either amplified or suppressed by trends in water clarity. *Limnology and Oceanography Letters* 1, 44–53.

Schmid M, Hunziker S, Wüest A. (2014) Lake surface temperatures in a changing climate: a global sensitivity analysis. *Climatic Change* 124: 301-315, doi:10.1007/s10584-014-1087-2

Stefan HG, Hondzo M, Fang X, Eaton JG, McCormick JH. (1996) Simulated long-term temperature and dissolved oxygen characteristics of lakes in the north-central United States and associated fish habitat limits. *Limnol Oceanogr* 41: 1124-1135, doi:10.4319/lo.1996.41.5.1124

Tanentzap AJ, Yan ND, Keller B, Girard R, Heneberry J, Gunn JM, Hamilton DP, Taylor DA. (2008) Cooling lakes while the world warms: Effects of forest regrowth and increased dissolved organic matter on the thermal regime of a temperate, urban lake. *Limnol Oceanogr* 53: 404-410, doi:10.4319/lo.2008.53.1.0404

Woolway, R. I., Merchant, C. J., Van Den Hoek, J., Azorin-Molina, C., Nöges, P., Laas, A., et al. (2019). Northern Hemisphere atmospheric stilling accelerates lake thermal responses to a warming world. *Geophysical Research Letters*, 46, 11,983-11,992. <https://doi.org/10.1029/2019GL082752>

J. Butcher

Controls of thermal response of temperate lakes to atmospheric warming

Jian Zhou, Peter R. Leavitt, Kevin C. Rose, Xiwen Wang, Yibo Zhang, Kun Shi, Boqiang Qin

Submission ID: NCOMMS-23-05303

REVIEWER COMMENTS

Reviewer #1 (Remarks to the Author):

General comments

This is a valuable paper, interesting, novel and generally well written and presented. I have some concerns about terminology and lack of quantification of statements, as well as other points below.

With due adjustment relative to the comments below, I will be happy to recommend publication.

Response: Thank you for your careful review and support of our manuscript. Your feedback is greatly appreciated and helpful, and as outlined below, we have addressed all points that you have raised, including use of a 'threshold', the relationship between ETT and HTT, and lack of clarity in terminology, quantification, and other unspecified points.

According to the reviewer's suggestions, we have made the following main actions:

1) We have reviewed all terminology in the manuscript for precision and accuracy. Vague or subjective terms have been clarified or removed.

2) We have revisited any statements lacking quantification or specific details and revised them to be clear and data-driven.

3) We have retained the plot of ETT-ATT vs. air temperature, but have used a generalized additive model analysis rather than a threshold-based analysis. Our main point remains: that rates of lake heating compared to the air decline as the atmosphere warms – that is, that lakes are less

responsive to additional atmospheric warming.. We have removed the discussion of specific thresholds from the revised manuscript.

4) We have redrawn the Fig. 2 with histograms and added the relationship between ETT and HTT.

We hope we have addressed your most significant concerns with these clarifications and revisions. Please do not hesitate to provide any follow up questions, comments, or suggestions you may have. Your input has been invaluable for improving this work. Thank you again for your time and for reviewing this manuscript.

Specific comments

1. Abstract -- many of the points could be better quantified in the abstract, such as the magnitude of the sensitivities (I assume in K/K) being discussed, what a "significant decrease" is quantitatively, etc. Also, I find the "threshold" language on sensitivity is not really justified. Such language suggests a step change in sensitivity, which is not credible from the data. Even though you have imposed a two-fit model on it with an inflection point, use of "threshold" to describe that inflection point is misleading, both because there is no step change in sensitivity, and because the step change in slope of sensitivity is a feature of your choice of fit rather than a feature of the data.

Response: We fully agree these changes have strengthened our abstract and the clarity of our work.

To address your specific concerns, we have added quantitative details and statistics throughout the manuscript.

We appreciate you pointing out the misleading use of "threshold" and the unjustified suggestion of a step change in 'sensitivity' (now 'responsiveness'). You accurately note our model imposed an inflection point not truly supported by the data. Upon reflection, we agree threshold was inappropriate

terminology and the relationship we showed between temperature and sensitivity was not fully supported. Accordingly, we have retained the relationship of ETT–ATT vs. air temperature, but have added an analysis with generalized additive models. We have removed discussion of the threshold from the manuscript, but have retained the conclusion that cooler lakes are more responsive to increased air temperatures. Changes have been made in Line **28–29**, **Line 140–144**, and **Figure 4**.

2. Line 43 -- What is long term? Quantify. Lakes integrate over the weather on a range of timescales that is very variable between lakes and can be just days, ie, not really "long term".

Response: Thank you for pointing this out. To address this, we have defined "long-term" as monthly-to-annual scale. We recognize that some authors are concerned about ‘heat-waves’ in lakes, however, our point is making that the high specific heat of water precludes rapid, extreme temperature changes such as seen in the air or on land. Changes have been made in **Line 46**.

3. Line 44 "slowly developing" -- quantify, as above. Ice can form overnight. Is that slow? Suggest reword to be clear.

Response: Thank you for catching our imprecise use of "slowly developing" to describe a range of lake processes (e.g., ice cover, stratification, surface temperature, evaporation, and water level). As you note, some processes like ice formation can occur quite rapidly, not "slowly." To address this, we have removed the term "slowly developing" in the revised manuscript. Changes have been made in **Line 47**.

4. Line 53 -- O'Reilly is a good reference, but it is pretty old (trends before 14 years ago -- a lot has

happened since then). There are annual updates on trends in lake temperature and air temperature in the annual BAMS state of the climate reports that could be a source of more contemporary statements, and probably other papers.

Response: Thank you for the suggestion to update our references regarding recent trends in lake and air temperatures. In the revised version, we now refer to Jane et al. 2021 which quantifies trends of air and lake water temperatures from 1980 to 2017 and forms the basis for our new analyses. Changes have been made in **Line 56–57**.

References

Jane, S. F. et al. Widespread deoxygenation of temperate lakes. *Nature* **594**(7861), 66-70 (2021).

5. Line 57 - Grammar correction needed for "apparently cooling".

Response: Thank you for pointing this out. We have corrected the grammar into “even including whole-lake cooling”. Changes have been made in **Line 59–60**.

6. line 58 - There is no mystery that not all lakes follow air temperature, because insolation, windiness, etc are well known to be important. So a better wording is "importance of accounting for the other factors" rather than "understanding".

Response: We thank the reviewer for this comment. We have modified the sentence according to the reviewer’s suggestion. Changes have been made in **Line 60–61**.

7. Line 62 - I agree with these additional factors discussed, but the other meteorological factors are omitted. There are trends in insolation, cloudiness (and therefore radiation budget) and wind in some

areas, and these modify lake temperature - air temperature relationships too. In fact -- you point this out later, but I think it should be noted together here.

Response: According to the review's suggestion, we have moved up our discussion of meteorological drivers so they are introduced together, rather than separately. Changes have been made in **Line 64–68**.

8. Line 78 --What is the definition of "lake temperatures" here? Is it annual average, summer, ...? What depth(s)? This is in the methods section later, but it needs to be stated here for those reading from start to finish in the order presented.

Response: As you note, without clarifying the temporal window, depths, or metrics being considered, this phrase is ambiguous for readers proceeding linearly through our work. Following the reviewer's suggestion, we detailed the definition of "lake temperature" and changed "lake temperatures" to "the responsiveness of lake surface and deep water temperatures to atmospheric warming during summer". Changes have been made in **Line 84–85**.

9. Line 82 -- Here you mention irradiance -- I think it should be mentioned a little earlier as per previous comment

Response: We agree with the reviewer and have moved the introduction of irradiance alongside to the discussion of meteorological factors. Changes have been made in **Line 64–68**.

10. Line 92 -- figure 2 is referenced before figure 1 which is non-standard.

Response: Thank you for pointing this out. We reorganized the order of figures in the revised

manuscript. Changes have been made in **Line 86**.

11. Figure 2 a and b -- there are 347 lakes, which can't give these smooth distributions. The visual impression can depend on the fit made to give these smooth distributions, and I would prefer to see histograms.

Response: Thank you for the feedback on Figure 2. We agree that adding histograms showing the distribution of trends would strengthen the visualization and interpretation of these data. We have replotted Figure 2 to include histograms displaying the frequency distribution to see not just the range of values, but how these trend data are distributed across lakes. This provides greater context for understanding the degree of coherence vs. variability in lake temperature and atmospheric responses. Revising Figure 2 with histograms has strengthened the communication value and scientific validity of this work. Changes have been made in **Figure 2**.

12. Figure 2d - I would be more interested in the HTT vs ETT plot than HTT vs ATT, or maybe as well as?

Response: Following the reviewer's suggestion, we have added a panel showing the relationship between ETT and HTT in Figure 2. This allows readers to directly see the degree of coherence between surface and deep lake temperature changes. Some lakes show close coupling, while others exhibit differential warming or cooling trends in the epilimnion vs. hypolimnion. Comparing surface- and deep-water warming provides insight into changes in stratification, mixing dynamics, and heat redistribution within lakes. Changes have been made in **Line 107** and **Figure 2**.

13. Line 104 -- again, a sentence framing the HTT changes vs ETT is just as valid to include. The variability of HTT vs ETT seems interesting to me.

Response: Thank you for your comments. We added the distribution of HTT with ETT in Figure 2 in the revised manuscript. Changes have been made in **Figure 2**.

14. Line 107 -- The difference in trend is defined as "sensitivity". I wouldn't like this word for this quantity to become established as I think it is confusing. A sensitivity is universally a differential dx/dy , not $dy(dx/dy - 1)$ which is your definition. You are calculating "trend differences", and I don't see a problem calling them trend differences. Always better to call something what it is rather than what it is not. For an example of why "sensitivity" is a bad name, consider a case, which could happen in your dataset, of a lake with ETT -0.1 K/dec and ATT $+0.1$ K/dec. In what sense does "a sensitivity of -0.2 K/dec" make sense: clearly the ETT in such a case is not driven by the ATT and is responding to other factors, rather than responding to ATT with a "negative sensitivity". Therefore please replace "sensitivity" with the more accurate "trend difference" throughout.

Response: Thank you for identifying our use of "sensitivity" as an inappropriate and misleading term to describe the difference between water and air temperature trends. You are quite right that sensitivity conventionally indicates a ratio (dx/dy) , rather than the difference $(dx - dy)$ we calculated. Using "trend difference" is a more accurate way to convey this metric. Following the reviewer's suggestion, we have removed the term "sensitivity" and used either "trend difference", "responsiveness", or "response" to discuss how the rate of lake heating differs from that of the atmosphere, particularly when we are considering how $dx-dy$ changes along a gradient of AT.

15. Line 109 -- "Analysis with random forest showed " is not very clear, please amend.

Response: Following the reviewer's comment, we revised the sentence to add an explanation of "Random forest analysis was used to determine which variables were most important in explaining temperature trend differences between air and water" in the revised manuscript. Changes have been made in **Line 125–126**.

16. Line 116 -- VPA has not been expanded as an acronym.

Response: Thank you for pointing this out. We modified the VPA abbreviation to "variance partitioning analysis" in the revised manuscript. Changes have been made in **Line 134–135**.

17. Fig 3 caption -- need to expand acronym MSE in the caption as this is read before the methods section.

Response: Thank you for pointing this out. We have expanded the acronym MSE in the caption of Figure 3. Changes have been made in the caption of **Figure 3**.

18. Line 124 -- seems to me to be more accurately phrased as "trend differences on average tended to be negative for $ATT > X$ deg C, and close to zero on average below that temperature". The reason is that your analysis shows many other factors are important and these may be confounding the effect: the warmer lakes warm slightly less fast than air temperature, but potentially because of your other influences and not because of a lower actual (local) sensitivity (in the dx/dy sense) to AT.

Response: Thank you for the thoughtful feedback. You make an excellent suggestion that we should pay attention to the distribution of $ETT-ATT$ with air temperature in relation to zero in Figure 4.

Therefore, we replace piecewise regression analysis with GAM analysis that includes a 95% credible interval and add a dashed line for null (zero) values. With the exception of the normal flair of the credible interval at the ends of the regression line, we see that ETT-ATT is significantly positive below $\sim 21^{\circ}\text{C}$, and negative above that value. However, as our intent is merely to show that there is a decline in the trend difference (aka response of lake to change in air temperature), we removed the narrative discussing the presence of a threshold. Here we focused on how the relationship of ETT-ATT changes with air temperature - the more complex compound effects will be evaluated from the perspective of lake heat balance in subsequent research. We appreciate you identifying this issue. Changes have been made in **Line 140-144** and **Figure 4**.

19. Line 138 -- this is the first time that it is mentioned that you are looking at "summer" air temperature -- needs to be clear before this. Including what "summer" means.

Response: Thank you for catching this oversight – it arises from defining the term only in the methods. We have now explicitly defined "summer" as July 15 to August 31 in the Northern Hemisphere, consistent with the peak open water season for most temperate lakes. Changes have been made in **Line 98**.

20. Line 138 -- the numbers are averages and there is variability. I think we need to have the fact these are means across 347 locations explicitly stated, and suggest to quote the variability (SD) as well as the mean values ($X \pm Y$).

Response: Thank you for pointing this out. We checked all the text, and added SD to all the mean values in the revised manuscript.

21. Line 168 -- the argument here seems to be that the land-use effect is mediated by altering the local air temperature -- but in that case why isn't all of that wrapped up in the SuAT factor, which is then correlated?

Response: Thank you for raising this insightful point. You are correct that if land use effects were mediated solely through changes in local air temperature, then summer air temperature (SuAT) should adequately capture that influence. However, as shown in Figure R1 below, land use also correlated significantly with other climate variables like solar radiation, wind speed, and spring air temperature in our study lakes.

For example, the percentage of agriculture and development were positively linked to SuAT and solar radiation, while forest cover was negatively related to wind speed, solar radiation, SuAT, and spring air temperature. Land use thus represents a more complex set of influences than air temperature alone. The patterns of land use also reflect the broad intensity and types of human activities within a watershed, encompassing elements beyond just impacts on air temperature. Land use affects lake ecosystems through multiple pathways, and should not be viewed as simply a proxy for local atmospheric warming. While air temperature is expected to partly mediate effects of land use on lake temperature, land use also correlates significantly with other climate drivers and more broadly represents human impact-factors which together help shape lake thermal regimes.

Figure R1 Pairwise correlations between land use and climatic variables are explored with Spearman’s correlation coefficient for the study lakes. The color gradient indicates the correlation coefficients (corr), and the square without cross indicate the correlations are significant ($P < 0.05$). Land use included agriculture, development, water, forest, wetland, grass, shrubland; climatic variables included summer wind speed (WS), total summer precipitation (TSP), shortwave radiation (SR), summer air temperatures (SuAT), spring air temperature (SpAT), fall air temperature (FaAT), and winter air temperature (WiAT).

22. Line 176 -- I think this speculation also needs a brief account of why you might think that -- e.g., the warm lakes are in locations of higher water vapour that inhibits the nighttime net radiative cooling of the lake.

Response: Thank you for catching our incomplete explanation and requesting clarification. We should have articulated the rationale for why warm lakes may lose more heat at night to support that speculation. As you note, warmer lakes would be expected to exhibit greater nighttime radiative cooling and heat loss to the atmosphere, all else being equal. This more rapid nighttime cooling could potentially counter daytime absorption, diminishing the net warming rate relative to the air

temperature in warm lakes and regions. We conducted further analysis and found that the decreasing responsiveness of surface WT to AT with increasing AT may be primarily related to changes in ETT, it is possible that diminished sensitivity to AT reflects arises because warm lakes may be more likely to lose latent heat to the atmosphere in temperate regions. It may explain why ETT–ATT significantly decreased with the increase of air temperatures in warm regions. Changes have been made in **Line 228–238**.

23. Line 189 -- I don't find this very convincing as a causal explanation. You would need to rule out that it arises from the sampling effect you then go on to discuss, I think, in order for it to be worth making this speculation. Can you rule out that it is all to do with confounding?

Response: Thank you for the insightful feedback. You make an excellent point that we cannot rule out confounding effects or sampling bias as the cause of the pattern we observed between lake temperature responsiveness and water temperature. As a result, we fully agree that speculation about the mechanism is unwarranted, and we have removed that discussion from the revised manuscript. Changes have been made in **Line 210–213**.

Rather than speculate, we intend to explore this relationship further in future work using more detailed data. For example, calculating full energy budgets for lakes across a climate gradient could provide better insights into how the thermal regimes and heat exchange dynamics of lakes vary systematically with temperature and other factors. More robust approaches are needed to support or refute our initial observation before hypothesizing mechanisms. Changes have been made in **Line 235–238**.

24. Line 210 -- You need to quantify the statements so that the size of this alleged reduction in sensitivity is clear. Looking at your plots, if a lake followed the red lines in figure 4, a lake that warmed from 24 degrees (at which is true sensitivity is on average 1 K/K) to 30 degrees would thereafter have a sensitivity of about 0.8 K/K. Another way to look at it is that 3 decades of mean summer T warming of 0.34 K/decade will on average reduce the sensitivity from ~1.0 K/K on average to ~0.965 K/K. This modest effect (which I don't even necessarily trust because I think you are seeing a confounding effect) should be matched by comparably modest wording of the finding. I find the current wording without quantification to give an exaggerated impression of importance. Please reword with appropriate quantification and appropriate explanation that the "reducing sensitivity" is at most a small effect compared to the lake warming itself.

Response: Thank you for the thoughtful feedback and prompting us to reexamine this issue in detail. In fact, air temperature itself reflects mixed effects because air temperature impacts not only itself but also many other influencing factors (e.g., humidity, wind speed), ultimately having a confounding effect on lake water temperature sensitivity. While global warming centers on air temperature, this paper primarily examines the relationship between ETT–ATT and air temperature (Spearman's correlation) as well as their change trends to provide insights into how lake water temperatures may shift under future global warming scenarios.

Linear regression analysis was used to explore the variation of ETT–ATT with air temperature changes. The results showed that the rate (slope) of ETT–ATT with air temperature changes was $-0.049 \pm 0.01^{\circ}\text{C}$ (Figure R2), meaning that if a lake's summer mean air temperature differs by 10°C , its trend differences may differ by 0.49 degrees. This rate of change is much higher than the 10-year warming trend for most lakes. Therefore, we do not think that the "reducing sensitivity"

(responsiveness) is at most a small effect compared to the lake warming itself.

Figure R2 The response of lake surface water temperature to air temperature changes varied with air temperature. The red line is a linear regression, and the grey area represents pointwise 95% confidence interval of the fitted values. Pairwise correlations between ETT-ATT and air temperature were examined by Spearman's correlation coefficient. ETT-ATT, the difference between ETT and ATT in summer.

25. Figure 4 and similar -- I suggest you need to add the zero line across the plot so we can see differences of the redline from zero more easily.

Response: Thank you for your suggestions. We have redrawn the Figure 4 and added the zero line across the plot in the revised manuscript. Changes have been made in **Figure 4**. We sincerely appreciate your comments, which have allowed us to significantly improve the manuscript.

Reviewer #2 (Remarks to the Author):**Key Results**

1. This paper presents a correlational analysis of thermal sensitivity of temperate lakes to atmospheric warming using information morphology and multi-decadal thermal profiles of 347 lakes. The paper is of interest because of the large data set assembled, which has not been common in the literature. Among the key points noted is that a majority of temperate lakes show increases in summer epilimnetic temperature over several decades, but greater than 50% of the studied lakes showed cooling trends in the hypolimnion. This is not a new finding and reflects complex interactions between onset of stratification, stratification strength, and seasonal weather patterns that have been predicted through modeling (e.g., Stefan et al., 1996), but it is nice to see this finding confirmed across an extensive data set. The paper also states that “sensitivity of surface WT to atmospheric heating decreased significantly when AT exceeded 20.2°C, suggesting a threshold for strong climate effects on lakes. Specifically, surface WT sensitivity to atmospheric warming may decline in a warmer future...” If true, this would constitute good news as a water temperature around 20 °C approximates a threshold for health of many cold water fish species. Unfortunately, I do not think the data fully supports this conclusion.

Response: Dear Dr. Butcher, thank you for your thoughtful review of our manuscript. We greatly appreciate the time and effort you invested to provide constructive feedback. Your comments have been enormously helpful in guiding us to reexamine our analyses and conclusions, identify weaknesses, and make substantive improvements.

According to the reviewer’s suggestions, we have made the following main actions:

1. The relationship of ETT–ATT with air temperature changes were retained, while the conclusion of a threshold in lake surface temperature sensitivity around 20°C, as claimed in the initial

manuscript, has been removed. Instead we focus on the decline in lake responsiveness to increased air temperature, and leave it to the reader to infer that there may be a threshold based on our GAM analysis (Fig. 4 and response to Reviewer 1).

2. For quality control, we matched metadata for all variables across datasets to calculate original trends in climate and lake temperature. Due to limited chlorophyll *a*, total nitrogen and total phosphorus data, meteorological and temperature data for some years were omitted in initial analyses. To accurately assess changes, we reorganized all data and filled gaps (in fact, nutrient data were only used in Supplementary Information Fig. S1) to maximize the complete monitoring record (expanded from 22.3 ± 7.5 to 24.5 ± 6.7 years). However, two outlier lakes (the trends of lake IDs 386 and 387 were -5.0 and -4.8°C per decade) were removed to avoid unduly influencing results.

3. To strengthening the explanatory power of our results, we added potential influencing factors, including water transparency, humidity, longwave radiation, regional sensible and latent heat fluxes, and area/depth ratio. The explanatory degree of ETT-ATT and HTT-ATT in random forest analysis increased and reached more than 30%.

4. Following the recommendations and references suggested by the reviewer, we fully modified the Introduction, hypotheses and much of the Discussion section.

We hope that with these changes, clarifications and a revised focus on more strongly supported conclusions, we have adequately addressed your concerns and feedback. The manuscript is much improved as a result of your diligent review. We appreciate you taking the time to thoughtfully evaluate our work and provide constructive comments to enhance scientific rigor.

Validity

2. The data analysis is based largely on application of random forest classification (followed by variation partitioning analysis and piecewise regression) to the difference between trends in water temperature and air temperature, which the authors refer to as “sensitivity.” Little information is provided on the calculation of trends except to state that they are based on Sen’s slopes. Sen’s non-parametric slope estimate is usually based on the median of slope estimates across multiple stations; it is apparently used here to represent a slope estimate at a given station over a given time period. There is no discussion of how these slope estimates may be biased by different start and end periods (as, for instance, might be expected by longer-term SST anomalies) for individual lakes.

Response: Thank you for your insightful comments. You raise an important point regarding potential biases introduced by differing start and end periods in the time series data. Indeed, differing start and end dates between lakes could influence the trend estimates (Figure R3 and Table R1). In fact, perfect data with consistent monitoring across all lakes is nearly impossible, especially given the challenges of *in situ* monitoring. We aimed to work within the constraints of available data, attempting to calculate consistent trend estimates for a given lake. Specifically, within any lake, we used consistent start dates and sampling frequency for the water and air temperature time series to facilitate comparing their trends.

Re-analysis of data from longer time series suggested that the more robust estimate of trend arose from longer time series (Table R1). For example, the trend of lake water temperature was stable from 1993-1996 to 2015, while the trend of water temperature from 1997-2001 to 2015 showed great differences (Table R1). Therefore, to better assess trends, we have supplemented meteorological and temperature data for years that were removed initially due to an absence of nutrient data (chlorophyll *a*, total nitrogen and total phosphorus). Consequently, all lakes used in this study have over increased

from 10 years to 15 years of monitoring data, with the total data span increased from 22.3 ± 7.5 to 24.5 ± 6.7 years.

In addition, the focus of our research is not quantifying the trends of climate and lake water temperature but rather to quantify differences in trends of water and air warming to estimate changes in lake response to air temperature change, as well as identify the primary correlated of these differences. Therefore, while the perfect dataset is not yet available, the monitoring programs used herein still provide invaluable insights into the state of lake systems over time. This work represents an initial exploration into factors linked to lake temperature sensitivity, but is intended to identify avenues for further, more detailed analyses, likely including detailed heat budgets. In this revision, we aim to accurately acknowledge the potential issues that you astutely identified when we interpreted the results, although we also recognizing that overcoming some challenges will rely on continued data collection and model development.

Figure R3 Distributions of air temperature trend (ATT), epilimnetic temperature trend (ETT), and their differences (ETT-ATT) with different start and end periods in Lake Washington and Lake

Maggiore. Raw data used in this figure detailed in below Table R1.

Table R1 The data of air temperature trend (ATT), epilimnetic temperature trend (ETT), and their differences (ETT–ATT) with different start and end periods in Lake Washington and Lake Maggiore.

Lake id	Lake	Start year	End year	ATT °C per decade	ETT °C per decade	ETT–ATT °C per decade
1	Washington	1993	2015	0.04672	0.00875	-0.03797
1	Washington	1994	2015	0.03543	-0.0075	-0.04293
1	Washington	1995	2015	0.04518	0.015	-0.03018
1	Washington	1996	2015	0.04518	0.02854	-0.01664
1	Washington	1997	2015	0.05964	0.0025	-0.05714
1	Washington	1998	2015	0.09036	0.06545	-0.02491
1	Washington	1999	2015	0.13263	0.08444	-0.04819
1	Washington	2000	2015	0.12684	0.06856	-0.05828
1	Washington	2001	2015	0.10678	0.06545	-0.04133
1	Washington	1993	2014	0.03526	0.00222	-0.03304
1	Washington	1993	2013	0.00902	-0.01533	-0.02435
1	Washington	1993	2012	-0.0169	-0.04	-0.0231
1	Washington	1993	2011	-0.01616	-0.044	-0.02784
1	Washington	1993	2010	0.01643	-0.04	-0.05643
1	Washington	1993	2009	0.03591	-0.04083	-0.07674
1	Washington	1993	2008	0.02851	-0.04575	-0.07426
1	Washington	1993	2007	0.05451	-0.03	-0.08451
6	Maggiore	1988	2015	0.03861	-0.02895	-0.06756
6	Maggiore	1989	2015	0.03619	-0.02357	-0.05977
6	Maggiore	1990	2015	0.03506	0.00647	-0.02858
6	Maggiore	1991	2015	0.04055	0.02587	-0.01468
6	Maggiore	1992	2015	0.05466	0.02357	-0.03109
6	Maggiore	1993	2015	0.07009	0.044	-0.02609
6	Maggiore	1994	2015	0.06469	0.03426	-0.03043
6	Maggiore	1995	2015	0.09346	0.05678	-0.03668
6	Maggiore	1988	2014	0.02357	-0.04273	-0.0663
6	Maggiore	1988	2013	0.03951	-0.02881	-0.06831
6	Maggiore	1988	2012	0.02779	-0.04553	-0.07332
6	Maggiore	1988	2011	0.02255	-0.05556	-0.07811
6	Maggiore	1988	2010	0.02728	-0.05154	-0.07882
6	Maggiore	1988	2009	0.01757	-0.06042	-0.07799
6	Maggiore	1988	2008	0.00418	-0.078	-0.08218
6	Maggiore	1988	2007	0.00694	-0.07429	-0.08123
6	Maggiore	1988	2006	0.02357	-0.05556	-0.07912
6	Maggiore	1988	2005	0.01064	-0.06943	-0.08007

3. The validity of the suggested threshold in response at AT of 20.2°C seems questionable. This

appears to be obtained from the breakpoint in the piecewise regression in Figure 4(a) (line 502) and is discussed at line 172 of the text. As noted at line 171-177, “effects of atmospheric warming on ET diminished when lakes were in warmer regions ($AT > 20.2^{\circ}\text{C}$), or were themselves already warm ($> 23.8^{\circ}\text{C}$). Because air temperature was correlated to trophic status...it is possible that diminished sensitivity to AT reflects arises because eutrophic lakes are often warmer than clear lakes... Resolution of the underlying mechanism will likely require comprehensive energy budgets, although we speculate that warm lakes may be more likely to lose heat to the atmosphere at night in temperate regions...” Figure 4(a) suggests the evidence for the breakpoint is weak while the discussion acknowledges that differences among climatic regions may also have a significant impact on the finding. Thus, I think it is inappropriate to bring the suggested breakpoint forward as a major finding in the Abstract.

In general, the validity of inferences presented in the paper is difficult to ascertain as the analysis is based on correlation or classification and not on causal hypotheses.

Response: Thank you for the insightful feedback identifying issues with the threshold conclusion and causal inferences in our work. You are correct that we lack definitive evidence of a discrete temperature threshold. Therefore, we have retained our analysis of the relationship of ETT–ATT with air temperature but have incorporated a GAM analysis, and have removed discussion of the breakpoint and threshold from the abstract and main text in the revised manuscript. As noted above, we see clear evidence that the lake responsiveness to atmospheric warming declines with air temperature (a central conclusion), but have not considered the trend in terms of a threshold response. For example, changes have been made in **Line 140–144** and **Figure 4**.

Continued data collection and model development is needed to address whether the relationship

between ETT–ATT and AT is non-linear of exhibits a sudden change. We aim for future work to build on these observational analyses with experimental and mechanistic studies probing how, why, and under what conditions lake temperature sensitivity changes. We sincerely appreciate your diligent feedback highlighting the need to clarify inferences and avoid overstating conclusions. Closing the gaps between correlation, causation and mechanisms is challenging but essential work. Your review has provided invaluable guidance to strengthen our science and communication. We will be sure to revisit your comments in revising and planning future analyses.

Significance

4. I am pleased that the authors have assembled and analyzed a large dataset of temperate lakes. While the analysis is relatively simple and does not break new ground, surveys across large datasets such as this are uncommon in the literature. I think, however, that the paper could be greatly improved by comparison to hypotheses of lake response to atmospheric warming obtained from simulation studies.

Response: Thank you for the positive feedback and helpful suggestions. Comparing observational results to modeling studies is an excellent way to provide context and evaluate consistency. According to the reviewer's suggestions, we have elaborated in the need to compare our results with key modeling studies and hypotheses around lake responses to atmospheric warming in both the Introduction and Discussion. Your suggestion has been enormously helpful in identifying ways to situate and convey our research that resonate with you and other readers. Changes have been made in **Line 65–79.**

Data and Methodology

5. This paper is a summary analysis across a large number of lakes, so I do not expect all data to be

provided with this paper or its supplemental information. The authors state that all data are available in references [44] and [45]. Can they affirm that this is true? I think it would be advisable to include summary information (particularly tabular information on Sen's slopes for each of the relevant variables) in the supplemental information for this paper.

Response: Thank you for requesting confirmation that the data are fully available, and suggesting the inclusion of summary tables in supplemental information. Providing transparent, accessible data is essential for science, and we appreciate you prompting us to examine this important aspect.

You are right that as a large survey analysis, this paper itself could not present raw data for all lakes studied. We now make it clear that the underlying data are available through published databases listed in Jane et al. 2020 and Stetler et al. 2021, both of which were co-authored by KR. We have also included basic summary information within the supplement to give readers a sense of the scope and ranges in factors like air and water temperature trends across lakes. In addition, following the reviewer's suggestions, we used the same methods and data source (the ERA-5 reanalysis), adding humidity, longwave radiation, regional sensible heat flux, and regional latent heat flux during summer in the revised manuscript. The data of climatic, water temperature, and water transparency trends used in this study are available as Supplementary data.

References

- Jane, S. F. et al. Widespread deoxygenation of temperate lakes: companion dataset 1980-2017 ver 1. Environmental Data Initiative, <https://doi.org/10.6073/pasta/ac8b05bb0da19032b3df3efc21f83874>, Accessed 2021-11-17 (2021).
- Stetler, J. T., Jane, S. F., Mincer, J. L., Sanders, M. N. & Rose, K. C. Long-term lake dissolved oxygen and temperature data, 1941-2018 ver 1. Environmental Data Initiative, <https://doi.org/10.6073/pasta/312f45d8d2ceaecf0c02e791f5fd9a63>, Accessed 2021-1-10 (2021).

Analytical Approach

6. The analytical approach consists of applying random forest analysis to the difference in trends (i.e., rate of change) in air temperature and water temperature, followed by variation partitioning analysis of groups of explanatory variables and piecewise linear regression. The potential explanatory variables include various measures of climate (e.g., summer air temperature, wind), lake morphology, and watershed characteristics. The analyses appear to be correctly implemented, although I would suggest supplying the R code in supplementary information.

Response: Thank you for the positive feedback on our analytical approach and implementation. We appreciate your suggestion to provide the R code used in our analysis. Your feedback has helped strengthen our data sharing practices, and ability to clearly communicate research methods. The source R code used in this study are publicly available at https://github.com/Laker-NIGLAS/Source_code. Changes have been made in **Line 357–358**.

7. While the relationships of trends in epilimnetic and hypolimnetic summer water temperatures to various explanatory variables is statistically significant, they only explain a small portion of the total variation with an unexplained residual of 80% for epilimnetic water temperature and 74% for hypolimnetic water temperature. Climate inputs have the greatest explanatory power, while, somewhat surprisingly, lake morphological characteristics did not emerge as a significant contributor to hypolimnetic temperature.

Response: Thank you for the thoughtful feedback highlighting uncertainties and limitations in the explanatory power of our analyses. You rightly note that about 80% of variation in surface and deep water temperature trends remains unexplained, indicating the complexity of dynamics driving lake

water temperature responses to climate warming. We appreciate you taking the time to call attention to this.

In this study, trend differences (e.g., ETT–ATT and HTT–ATT) are an indirect response variable which introduces additional variability not seen when directly analyzing ETT and HTT. Noted that over 50% of variation in ETT alone was explained by the factors explored. This helps articulate why relationships for ETT–ATT, while significant, explained less overall variation.

Solar radiation is the primary energy source determining lake heat budgets, so climate factors like irradiance, air temperature, wind, and humidity are overriding controls on differences in atmospheric and lacustrine warming rates. In contrast, factors redistributing heat within lakes likely have secondary effects. We were somewhat surprised morphological characteristics did not strongly relate to hypolimnetic temperature trends. However, during summer stratification, lake morphometry may have limited influence on deeper waters, instead mainly affecting the epilimnion. In fact, when lake stratified, hypolimnetic temperatures are predominantly controlled by heat diffusion from above water rather than by lake shape or depth directly. At these times, surface area, location and water renewal largely govern epilimnetic conditions alone. Our results align with this idea, suggesting morphological characteristics shape hypolimnetic temperature trends indirectly by influencing surface heat budgets and the degree of stratification. Complex interactions with other factors like water clarity can also obscure simple morphometric relationships.

In the revised manuscript, we have made the following changes:

1. For quality control, we matched metadata for all variables across datasets to calculate original trends in climate and lake temperature. Due to limited chlorophyll *a*, total nitrogen and total phosphorus data, meteorological and temperature data for some years were omitted in initial analyses.

To accurately assess changes, we reorganized all data and filled gaps (in fact, nutrient data were ultimately unused in Supplementary Information Fig. S1) to maximize the complete monitoring record (expanded from 22.3 ± 7.5 to 24.5 ± 6.7 years).

2. To strengthening the explanatory power of our results, we added potential influencing factors, including water transparency, humidity, longwave radiation, regional sensible and latent heat fluxes, and area/depth ratio. The explanatory degree of ETT-ATT and HTT-ATT in random forest analysis increased and reached more than 30%.

3. Remove outlier lakes. Two outlier lakes (the trends of IDs 386 and 387 were -5.0 and -4.8°C per decade) were removed to avoid unduly influencing results.

Your feedback has helped us think critically about our results and discussion, ensuring we properly represent the complexity and uncertainties at play. Highlighting large unexplained variation and possible reasons for weaker relationships prompts us to consider interactions and dynamics beyond what we analyzed. The specific comments you provided guide us to address meaningful limitations that shape how reader might interpret conclusions, helping avoid overstating implications.

8. The variation analysis is a useful exploratory tool, but can only evaluate information that is supplied to it. The low amount of explained variation may reflect the complexity of interactions between various inputs or the lack of information on key factors. For instance, water clarity may play an important role in the relative temperature trends in surface and bottom waters, but water clarity is not an explanatory variable in the variation analysis (despite Secchi depth being used in other parts of the paper), although it is likely correlated with some of the land use variables. For lake morphology, surface area and depth taken individually are not significant predictors, but it is more likely the ratio of these variables that is a significant influence on mixing regime.

Response: Thank you for the insightful feedback on limitations and opportunities to strengthen our variation analysis. You point out that the explanatory power depends entirely on which variables are included, and call our attention to potentially important factors our initial analysis lacked, such as water clarity and lake morphometry metrics like the ratio of area to depth. We appreciate you taking the time to suggest specific ways we could improve this work and better understand trends.

In the revised manuscript, we have:

1) Added Secchi depth (water clarity) as an explanatory variable in the random forest and variation partitioning analyses. Despite being correlated with land use, water clarity itself likely contributes to the relative degree of surface vs. bottom warming in lakes.

2) Included the ratio of lake surface area to maximum depth as an additional morphological variable. This integrates surface area and depth in a way directly relevant for understanding a lake's propensity for mixing and stratification.

3) Added several other potential climatic drivers including humidity, regional sensible heat flux, and regional latent heat flux. Recognizing the complexity of interactions influencing lake temperature change, we aimed to represent more factors that could impact warming or mediate responses.

Your feedback has helped us think more broadly about relationships and variables to explore that could shape how lakes are responding to climate change. Adding more potential drivers and discussing limitations around what we could include helps avoid missing key factors, and prompts consideration of next steps for developing understanding.

9. The piecewise linear regression is used to propose a breakpoint in the response of water temperature to air temperature, which seems suspect as noted above. Additional statistical analysis is needed on this point to determine whether the apparent change in response slope at the breakpoint is actually

significant.

Response: Thank you for the thoughtful feedback. Indeed, it seems suspect to propose a definitive threshold temperature. Therefore, we replace piecewise regression analysis with GAM analysis to show the relationship of ETT–ATT with air temperature changes. In the revised version, we have removed discussion of the breakpoint and threshold from the abstract and main text, and focused on the trend of ETT–ATT as a function of air temperature. For example, changes have been made in **Line 140–144** and **Figure 4**.

Suggested Improvements

10. The paper suggests relationships between trends in atmospheric temperature and lake water temperature in both the epilimnion and hypolimnion of temperate lakes. However, the analysis is primarily correlational in nature and presents (or perhaps confirms) findings that have already been suggested by others. The paper could be greatly improved by reframing the analyses as an evaluation or test of hypotheses that have been proposed in the existing literature.

Response: Thank you for the insightful feedback on reframing our analysis to evaluate existing hypotheses - doing so strengthens the scientific integrity and impact of our work. Reframing the analysis around established hypotheses gives our work purpose and context, strengthening both scientific value and communication. Your recommendation addresses what was missing in our initial draft. We sincerely appreciate you taking the time to provide thoughtful feedback identifying this key opportunity for improvement. Changes have been made in **Line 87–90**.

11. Many of the responses evaluated here have been investigated previously through model experiments related to climate change. It would be useful to compare and contrast some of these

inferences to the results reported here. Some of these findings include the following:

- Lake surface area and depth (and especially their ratio) affect strength of stratification and can result in a net decrease in whole lake average temperature (Stefan et al., 1996; Tanentzap et al., 2008).
- Increased water clarity can warm deeper waters; conversely reduced clarity can reduce the depth of light penetration as well as the mixing depth and reduce hypolimnetic temperatures, although responses in individual lakes are complex (Stefan et al., 1996; Hocking and Straškraba 1999; Rose et al., 2019).
- In addition to effects of lake morphometry, anomalies in the responses of individual lakes to increased atmospheric temperature is modulated by changes in relative humidity and wind speed (Schmid et al., 2014; Woolway et al., 2019).

Response: Thank you for recommending we compare our results to inferences and findings from previous modeling experiments related to climate change impacts on lakes. Doing so helps contextualize our work, identify consistencies and inconsistencies, and spur new hypotheses or areas of needed research. We appreciate you highlighting this opportunity to strengthen interpretation and discussion. In the revised manuscript, we have added comparison of our results to the relevant studies you mentioned that propose or explore relationships through modeling. For example, changes have been made in **Line 69–75**.

Clarity and Context

12. The writing is generally good; however, organization of the paper could be improved. As currently structured, the paper contains two major section heads (Results, and Methods) with no subheads. The Nature style is to put details of Methods after Results; however, it would be helpful to put somewhat

more summary information on methods in the Results section if it is presented first. In addition, the long Results section should be divided with subheads to aid the reader.

Response: Thank you for the feedback on improving organization and clarity within our manuscript. Grouping results into subsections, providing more methodological context within the results, and reorganizing the methods section are excellent suggestions.

In the revised manuscript, the results section has been divided into three subsections:

- Long-term variations of air and lake water temperatures
- Controls of the responsiveness of lake water temperature to air temperature
- Effects trophic status on lake responsiveness to atmospheric warming

This grouping collects similar analyses and aids reader navigation. Subheadings provide an overview of what is covered in each part. In addition, we have provided more methodological context within the results. For example, we added an introduction to random forest analysis (**Line 125–126**) and clarification of criteria for lake trophic status (**Line 146–147**).

References

13. The paper contains a relatively comprehensive survey of relevant literature. The authors should consider also citing the following (see below for complete citations):

- Hocking and Straškraba, 1999
- Rose et al., 2016
- Tanentzap et al., 2008
- Woolway et al., 2019

In addition, Pilla et al. (2020) is mentioned only in passing but likely deserves a bit more discussion.

Response: Thank you for recommending additional references relevant to our work. Incorporating

insights and ideas from a diversity of studies strengthens scientific work by providing context and identifying open questions. We appreciate you taking the time to suggest specific papers that would benefit our analysis and discussion. According to the reviewer's suggestions, we carefully read all the references you mentioned, and introduced and cited them in the revised manuscript.

Thank you again for recommending these resources - doing so made incorporating insights from each work straight-forward and ensured we considered them thoughtfully. Your time is sincerely appreciated.

REVIEWERS' COMMENTS

Reviewer #1 (Remarks to the Author):

The authors have responded to all comments thoughtfully and appropriately, and I am happy to recommend publication.

Reviewer #2 (Remarks to the Author):

The authors have responded in detail to my comments as well as to those of Reviewer 1. I had already felt that the work was a useful addition to the literature on lake response to climate change, but pointed out several areas in which greater clarity or details were required in the presentation. The authors have been attentive to those requests and have substantially revised the paper in the ways recommended by both reviewers. This includes comments relative to results, significance, conclusions, data analysis, and methodology. In addition, the authors have clarified and provided links to the original data used in their analysis. I continue to believe that the research is not groundbreaking but is a useful synopsis of a large data set and has been reframed to test the hypotheses proposed by others on less comprehensive data. Given the substantial improvements and clarifications in the revised version it is my opinion that the article is now appropriate for publication.

Author's response to reviewers' comments

Controls of thermal response of temperate lakes to atmospheric warming

Jian Zhou, Peter R. Leavitt, Kevin C. Rose, Xiwen Wang, Yibo Zhang, Kun Shi, Boqiang Qin

Submission ID: NCOMMS-23-05303A

REVIEWER COMMENTS

Reviewer #1 (Remarks to the Author):

The authors have responded to all comments thoughtfully and appropriately, and I am happy to recommend publication.

Response: We greatly appreciate the time and effort that the reviewer has dedicated to providing thoughtful feedback on our manuscript. We are pleased to know that the reviewer is satisfied with our responses and revisions, and finds them appropriate. It is encouraging to receive such a positive endorsement, and it motivates us to continue conducting rigorous research. We are thankful that the reviewer recognizes the care we have taken to address all comments thoroughly. With the reviewer's approval, we look forward to the publication process and sharing our work with the scientific community. We hope that this research will make a meaningful contribution to the field. Thank you again for your time and the opportunity to improve our manuscript through this peer review process.

Reviewer #2 (Remarks to the Author):

The authors have responded in detail to my comments as well as to those of Reviewer 1. I had already felt that the work was a useful addition to the literature on lake response to climate change, but pointed out several areas in which greater clarity or details were required in the presentation. The authors have been attentive to those requests and have substantially revised the paper in the ways recommended by both reviewers. This includes comments relative to results, significance, conclusions, data analysis, and methodology. In addition, the authors have clarified and provided links to the original data used in their analysis. I continue to believe that the research is not groundbreaking but is a useful synopsis of a large data set and has been reframed to test the hypotheses proposed by others on less comprehensive data. Given the substantial improvements and clarifications in the revised version it is my opinion that the article is now appropriate for publication.

Response: Dear Dr. Butcher, we sincerely appreciate the time and effort you have dedicated to providing constructive feedback on our manuscript. We are thankful that you feel our work makes a useful addition to the literature on lake response to climate change. We are pleased that you acknowledge we have been attentive to the requests for revisions, and find that the manuscript has been substantially improved in the recommended ways. We look forward to sharing the published work with the scientific community.